# Functional Intelligence-Based Scene Recognition Scheme for MAV Environment-Adaptive Navigation

**Lingling Wang** [1], **Yixin Liu** [1] , **Li Fu** [1,*], **Yaning Wang** [2] and **Ning Tang** [3]

1   National (Virtual Simulation) Experimental Teaching Demonstration Center of Mechanical and Control Engineering, Beihang University, Beijing 100191, China; wangling0908@buaa.edu.cn (L.W.); sasee_lyx@buaa.edu.cn (Y.L.)
2   North Automatic Control Technology Institute, Taiyuan 030000, China; wyn_ning@buaa.edu.cn
3   Computer Science and Control Systems, Bauman Moscow State Technical University, Moscow 105005, Russia; tangning@buaa.edu.cn
*   Correspondence: fuli@buaa.edu.cn; Tel.: +86-10-8231-7306

**Abstract:** Adaptive navigation is the core of micro aerial vehicles (MAVs) conducting autonomous flights in diverse environments. Different navigation techniques are adopted according to the availability of navigation signals in the environment. MAVs must navigate using scene recognition technology to ensure the continuity and reliability of the flight. Therefore, our work investigated the scene recognition method for MAV environment-adaptive navigation. First, we exploited the functional intelligence-adaptive navigation (FIAN) scheme by imitating the physiological decision-making process. Then, based on sufficient environment-sensitive measurements from the environment perception subsystem in FIAN, the two-level scene recognition method (TSRM) in the decision-making subsystem consisting of two deep learning frameworks, SceneNet and Mobile Net-V2 was proposed to extract scene features for accurate diverse scenes recognition. Furthermore, the four-rotor MAV-Smartphone combined (MSC) platform simulating the owl's omni-directional head-turning behavior was built. The proposed TSRM was evaluated for accuracy, delay, and robustness compared with PSO-SVM and GIST-SVM. The results of practical flight tests through MSC platform show that TSRM has higher classification accuracy than PSO-SVM and GIST-SVM, and performs smoothly with self-regulatory adaptations under diverse environments.

**Keywords:** decision-making system; functional intelligence; scene recognition; MAV-smartphone combined platform; deep learning

## 1. Introduction

MAVs have superior agility and hover capability, making them ideal for surveillance, search and rescue missions in various environments. Inertial sensors insusceptible to environmental changes are ideal sensors for MAVs. However, most MAVs are equipped with low-cost inertial MEMS sensors that lack long-distance navigation accuracy due to cost and payload constraints. A lot of work has been done to improve the navigation accuracy of MAVs under particular environments, such as visual-inertial navigation systems in indoor scenes [1–7] or well-developed GPS-inertial navigation systems in outdoor scenes [8–12]. However, exteroceptive sensors such as GPS and cameras that perform well under certain environmental conditions are prone to failure when the environment changes [13–15]. Therefore, one major challenge in autonomous flight is finding a way for MAVs to achieve satisfactory navigation accuracy in diverse environments.

Recently, researchers have investigated the environment-adaptive navigation problem for MAVs in changing environments. Although environment-adaptive navigation is a popular research topic, challenges remain when recognizing the environmental context precisely to reconfigure multisensor navigation systems in large-scale environments involving scene transitions, especially for high dynamic MAVs. A modular multisensor integrated

navigation system was proposed to adapt to different environments without redesigning the whole system [16]. A variable structure navigation system with high accuracy and fault tolerance was developed based on intelligent adaptive behavior [17]. However, the criterion for measuring sensors based on classifying the system's observability is susceptible to disturbances. Gao [18] came up with a hybrid environmental context classification scheme to classify indoor, intermediate, urban, and open-sky scenes based on the GPS signal from smartphones under different kinds of environments. The scheme obtained 88.2% accuracy. However, the test results in intermediate scenes are not satisfactory. Furthermore, the hybrid context detection method was not validated under high dynamic conditions, so it should be improved to work on MAVs.

The features extracted from smartphone sensor information will help us to improve the reliability of environmental detection. However, the time-series measurements of multiple sensors on the smartphone are affected by unmodeled time-varying noises in practice, and those noises are difficult to filter out with traditional techniques. To address this issue, Chen [19] proposed an ensemble extreme learning machine to extract the features of human activity from smartphone sensor data that are affected by noises. Yao [20] presented a unified deep learning framework that integrated convolutional and recurrent neural networks to exploit different types of relationships in smartphone sensor data with unmodeled noises and extracted distinct features effectively. Ding [21] came up with a deep neural network coined the noise-resistant network. Zhang [22] proposed an asymmetric encoder–decoder structure based on ResNet. Even though the effectiveness of the feature extraction was demonstrated to significantly outperform the state-of-the-art methods in low dynamic applications, these deep neural network-based scene recognitions need to be redesigned to work in high dynamic environments [23].

This paper considers the critical components of adaptive navigation for MAV to perform autonomous flights in diverse environments. Inspired by the functional systems theory [24], we proposed a functional intelligence-adaptive navigation (FIAN) scheme for MAVs. Unlike other navigation methods, the functional intelligent navigation method focuses on mimicking the decision-making and feedback mechanisms in biological cognitive processes, thus enabling active adaptation to the environment. The FIAN scheme mimics the biological characteristics of the owls' head twisting in all directions and visual-auditory conversion during hunting. The scheme allows the navigation process to perform as smoothly as the functional physiological systems with self-regulatory adaptations performed under diverse environments. A four-rotor MAV-smartphone combined (MSC) platform has been designed to develop the FIAN scheme. A two-level scene recognition method (TSRM) is exploited by mimicking the physiological decision-making process for the MAV to recognize information with accuracy. The environmental features are extracted from low-cost environment-sensitive sensors in a smartphone and used for scene classification in two deep learning frameworks, SceneNet and MobileNet-v2. The paper shows that the proposed scene recognition method is feasible to implement on a smartphone installed on the MSC platform. The scene recognition results have been verified through actual flight tests.

The paper is organized as follows: Section 2 describes the FIAN scheme and the design of the MSC platform. Section 3 analyses the smartphone's sensor data under different scenes and describes the two-level scene recognition method (TSRM) in detail. Section 4 introduces the processing and training methods for SceneNet and MobileNet-v2. Section 5 assesses the flight's performance, which shows the effectiveness of the proposed method. Lastly, in Section 6, we present the conclusions and topics for further research.

## 2. FIAN Scheme and MSC Platform

### 2.1. FIAN Scheme

Functional systems are distinguished from reflex systems in which information propagates linearly from receptors to executive organs via the central nervous system. Instead, functional systems are various anatomical systems with self-organized nonlinear character-

istics to describe the behavioral structure of organisms and perform diverse functions, such as breathing, swallowing, locomotion, etc. [24,25]. For example, the owls' visual-auditory conversion behavior in hunting clearly illustrates the mechanism of functional systems. First, the owls turn their heads in all directions to find prey through vision. When the field of view is blocked, owls will switch to use their auditory to detect the prey location. The intelligent control systems are synthesized based on the functional systems theory and have shown remarkable adaptability to the change of environment [20]. Inspired by the functional systems mechanism of owls, we developed a functional intelligence-adaptive navigation (FIAN) scheme for MAVs shown in Figure 1.

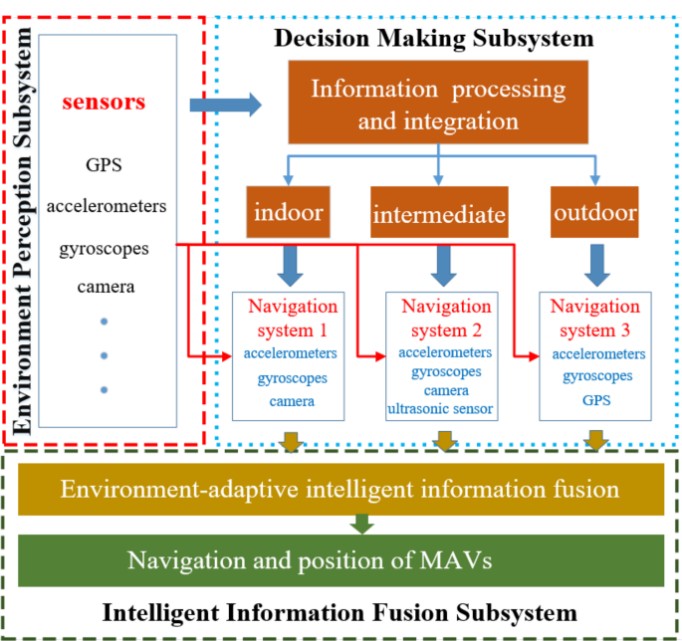

**Figure 1.** FIAN scheme for MAVs.

The scheme combines three subsystems to address the environment-adaptive navigation problems: the environment perception, decision-making, and intelligent information fusion subsystem. The environment perception subsystem comprises multiple sensors that provide environmental and behavioral information for the decision-making subsystem. The decision-making subsystem processes environment-sensitive data to classify flight scenes, and it guides the intelligent information fusion subsystem to reconfigure the multi-sensor navigation system, thus enabling the fusion of information from different navigation systems.

The function of the decision-making subsystem is to identify the current flight scenes of the MAV, which is the key to achieving the environment-adaptive navigation. This subsystem enables the MAV to autonomously select the available information and improve the accuracy of navigation. For example, when MAV is flying from an outdoor scene to the interior of a building, the scene transitions during the flight will affect the signal availability of environment-sensitive sensors. Similarly, the tree cover will also affect the sensitivity of the GPS, camera and other sensors in different areas as the MAV is flying in the forest. The availability characteristics of various sensors mounted on MAV in a sparse forest area are similar to those in outdoor scenes. When the MAV flies in a dense forest area, some environment-sensitive sensors such as the GPS signal can be obstructed like in an indoor scene. In between sparse and dense areas, it is like an intermediate scene while traversing buildings. It can be seen that the scene properties in the forest have resemblance to those in urban building areas. Therefore, the decision-making subsystem divides the flight scenes into three categories: outdoor scenes, intermediate scenes, and indoor scenes.

When the MAV is in outdoor scenes such as playgrounds or sparse forest, the GPS navigation and positioning accuracy will be more precise and the GPS can integrate with

the inertial navigation system. While the MAV traverses the playground into intermediate scenes such as corridors, the multipath effect resulting in large pseudo-range errors of GPS will occur due to the satellite signal reflection caused by surrounding buildings [26]. At the same time, the change in light brightness due to the scene transitions makes the camera signal unstable. Considering that the ultrasonic sensors determine the distance via acoustic reflection, MAVs in the intermediate scenes achieve precise navigation by fusing ultrasonic and inertial sensor measurements [27]. After the MAV traverses the corridor and enters the indoor scenes, the GPS signal will not be able to meet the MAV navigation requirements due to obstructions [28]. At the same time, the complex barriers in the indoor scenes lead to the ultrasonic sensor measurements being unavailable. Therefore, the FIAN scheme can only use the visual/inertial integrated navigation system for MAV in the indoor scenes. After identifying the current flight scenes of MAV, the intelligent information fusion subsystem will continuously adjust the combination of sensor outputs according to the availability of sensor measurements in various scenes.

## 2.2. MSC Flight Platform Configuration

The platform for FIAN should be able to implement FIAN software without changing the hardware components. FIAN was also designed to fly under diverse environments, and is simple to secure operational reliability. Since smartphones have abundant environmental sensors and robust information processing capability, we use a smartphone as an environment perception and decision-making subsystem. Furthermore, a four-rotor MAV-smartphone combined (MSC) platform has been designed shown in Figure 2.

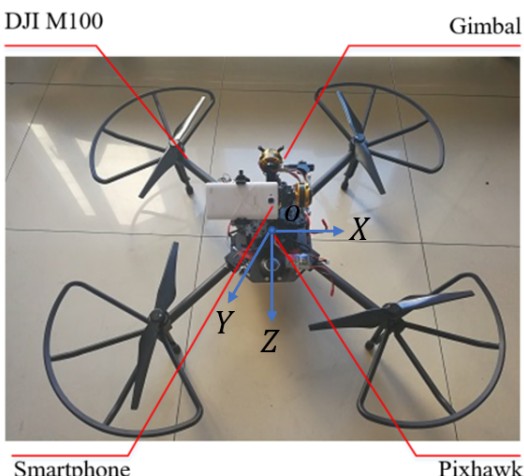

**Figure 2.** The configuration and coordinates of the MSC flight platform.

The MSC system mimics the owl's biological ability to rotate its head in all directions to observe the surrounding environment. Therefore, the smartphone is attached to a three-axis gimbal on the MAV, which allows the decision-making subsystem to obtain sufficient information about the environment. The MAV airframe DJI M100 is controlled by Pixhawk autopilot. The PX4-based system implements the intelligent information fusion subsystem to fuse the outputs from the inertial measurement unit, magnetometer, GPS, barometer, airspeed sensor, and an optical flow sensor in the Pixhawk autopilot. The smartphone communicates with the Pixhawk autopilot via the WIFI module ESP8266. The PX4-based system receives the decision-making subsystem in the smartphone to reconfigure the multisensor navigation system and forwards the gimbal commands to the storm32bgc controller via serial port.

## 3. A Two-Level Scene Recognition Method (TSRM)

### 3.1. Environment Perception Subsystem

The environment perception subsystem in the FIAN scheme provides sufficient environment-sensitive measurements to allow the decision-making subsystem to achieve accurate scene recognition results. Five environment-sensitive sensors in the smartphone are tested to analyze the features of different environments.

The tri-axial magnetometer is mounted pointing in the same direction as the tri-axial accelerometer in the smartphone on an MAV-fixed body coordinate, as shown in Figure 2. The coordinate origin is the center of gravity of the MSC, the X-axis points to the forward direction of the smartphone, the Y-axis points to the right, and the Z-axis is determined by the right-hand rule. Figure 3 illustrates the three-axis magnetometer outputs under different scenes. The X-axis, Y-axis, and Z-axis magnetometer outputs in indoor scenes are affected by complex electromagnetic interference and ferromagnetic substance in reinforced concrete buildings. The outputs also demonstrate that there is more significant variance of measurements in indoor scenes than that in the outdoor scenes, while the measurements of the intermediate scenes are in between the indoor scenes' measurements and outdoor scenes' measurements.

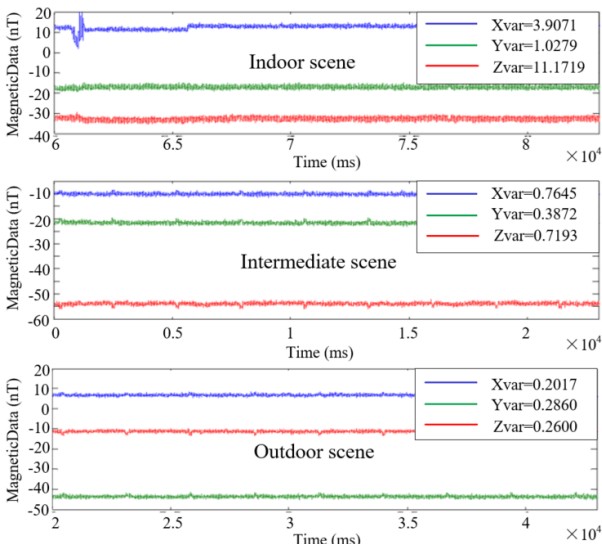

**Figure 3.** The magnetometer measurements in three different environments.

The primary light source is sunlight in the outdoor scenes and artificial light or the reflection of sunlight in the indoor scenes. Therefore, the light intensity in indoor scenes is significantly lower than in outdoor scenes, and the light intensity in intermediate scenes is in between, as shown in Figure 4.

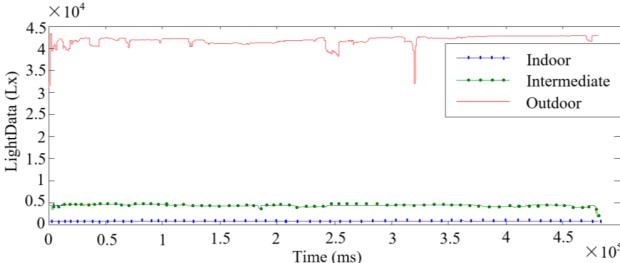

**Figure 4.** The measurement of light intensity sensors in three different environments.

The position accuracy of GPS depends on the signal strength and the number of visible satellites. Figure 5 shows the GPS signal strength of visible satellites in indoor, outdoor, and

intermediate scenes tested by AndroiTS GPS in the smartphone. The number and signal strength of visible satellites are significantly different in diverse environments.

Figure 5 indicates the quality of the acquired satellite signals in green, yellow, and grey. The color coded as green denotes that high precision navigation results can be obtained using these signals with strong strength. When the low quality satellite signals in yellow are used, the navigation and position accuracy of the GPS will be reduced. The grey signals cannot be used in GPS navigation. In the outdoor scenes shown in Figure 5c, partially available satellite signals such as No.15 and No.24 with good quality can be acquired. In the cases of the intermediate and indoor scenes shown in Figure 5a,b, some satellite signals can be received, but none of these signals can be used for navigation and positioning. The above analysis shows that relying solely on GPS signals cannot identify the current flight scenes of the MAV in real-time, because it does not consider the differences in amplitude that the data may have [29].

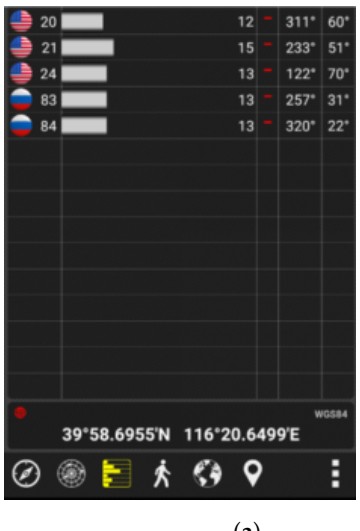

(**a**)

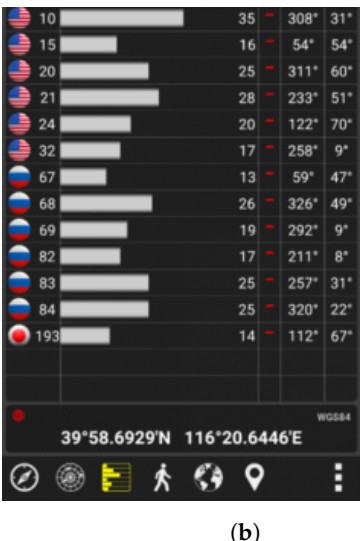

(**b**)

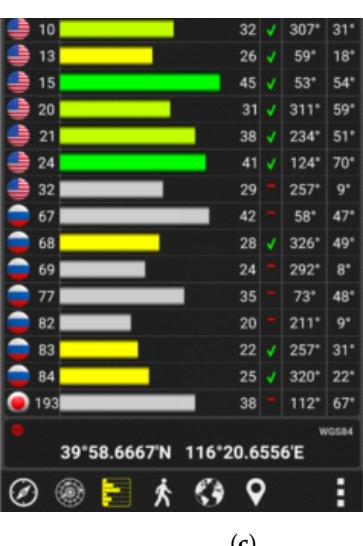

(**c**)

**Figure 5.** The status of visible satellites in three different environments. (**a**) Indoor. (**b**) Intermediate. (**c**) Outdoor.

The barometer in the smartphone can determine the altitude according to atmospheric pressure measurements. When an MAV is flying in an enclosed indoor space, the flight altitude and vertical speed are usually restricted, but they are unrestricted when the MAV flies in an open outdoor scene. This information can be used to improve the accuracy of scene recognition.

Various features in images could contribute to identifying the environment of the MAV as well. Outdoor scenes tend to have more smooth lines and fewer edges in the image. Intermediate scenes have a medium proportion of sky and some buildings. Indoor scenes have a low proportion of sky and more edges.

### 3.2. The Two-Level Scene Recognition Method (TSRM) in the Decision-Making Subsystem

In the FIAN scheme, the main task of the decision-making subsystem is to accurately identify the current scene. Research shows that owls use their vision to observe their environment or sense the location of prey through hearing while hunting [30]. Inspired by the hunting behaviour of owls in complex environments, we exploited a two-level scene recognition method (TSRM) for the FIAN scheme, consisting of real-time recognition segment and triggered recognition segment, as shown in Figure 6. The decision subsystem can make use of different sensors to acquire navigation information according to diverse scenes.

During MAV flight, the real-time recognition segment applies a deep learning network called 'SceneNet' to extract the environmental features from the measurements by GPS, magnetometer, light intensity sensor, and barometer to recognize the scene with a high

update rate. The function of 'Belief' is to obtain the classification confidence probability of indoor scenes, intermediate scenes and outdoor scenes from SceneNet, and take the maximum value of these three confidence probabilities for further scene recognition. When the maximum probability is larger than the threshold value, it means that the scene recognition rate of the real-time recognition segment is high and 'Belief' will export the recognized scene information to the intelligent information fusion subsystem of FIAN scheme. While the belief of the real-time recognition segment is below the trigger threshold, the MAV transits to hovering mode, and the triggered recognition segment will be activated. The triggered recognition segment generates five gimbal commands to acquire enough information from the gimbal angles $(0°, 0°, 0°)$, $(0°, 0°, 90°)$, $(0°, 0°, 180°)$, $(0°, 0°, 270)$ and $(90°, 0°, 0°)$, and then MobileNet-v2 network is applied to obtain the result.

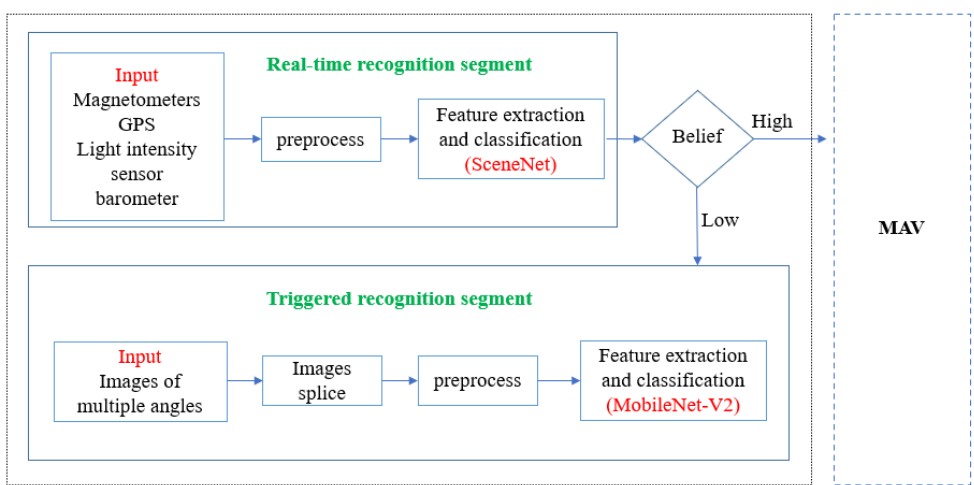

**Figure 6.** The two-level scene recognition method (TSRM).

### 3.3. Real-Time Recognition Segment

A new deep learning network, "SceneNet", t5 is exploited in the real-time recognition segment, as shown in Figure 7. This segment aims to use measurements of non-vision-based sensors in a smartphone to achieve environment feature extraction and classification.

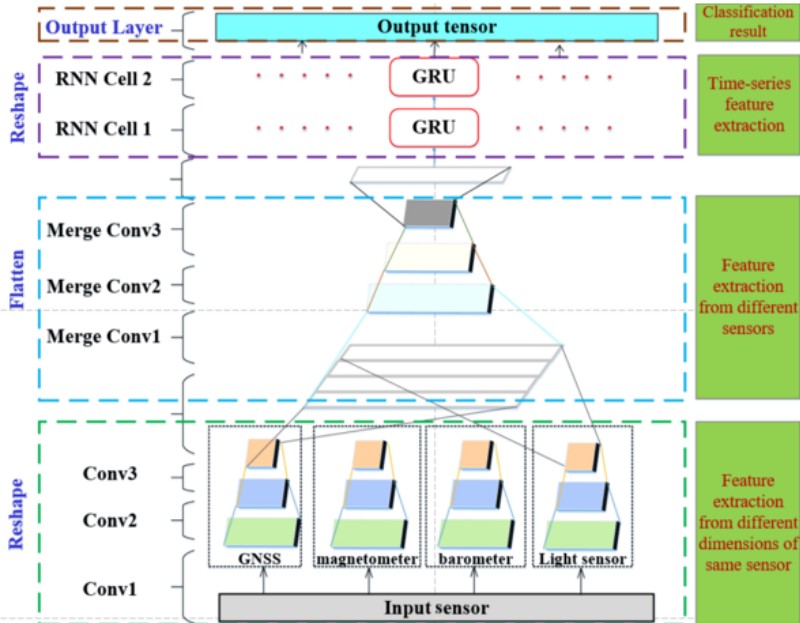

**Figure 7.** The architecture of "SceneNet" in real-time recognition level.



At the reshape process, four convolutional neural subnetworks are constructed to exploit internal relationships between different measurement dimensions of the same sensor. Each convolutional neural subnetwork contains three individual convolutional layers with different kernel sizes. The flatten process consists of three convolutional neural layers with different kernel sizes to merge the results of the reshape process and find the relationships between the data of different sensors. The activation function in convolutional layers is the RELU activation function shown below,

$$\text{RELU}(x) = \begin{cases} x, & x > 0 \\ 0, & x \leq 0 \end{cases} \tag{1}$$

where $x$ is the batch normalized input.

During MAV flights, the measurements of non-vision-based sensors in the smartphone change over time. The merged results have time series characteristics. Recurrent neural networks (RNN) can capture features from time-series data. The gated recurrent unit (GRU) [31] is a unique structure of RNN that can be used effectively to learn the implicit features in sequences with less cost. Two GRUs with dropout and batch normalized inputs are stacked on the framework to learn advanced features in diverse environments, as shown in Figure 7. In each GRU, 200 hidden neurons are set to memorize time-related states.

A fully connected layer with softmax activation function in Equation (2) is stacked on RNN layers.

$$P(i) = \frac{\exp(a_i)}{\sum_{k=1}^{T} \exp(a_k)} \tag{2}$$

where $a_i$ and $a_k$ are the $i$th and $k$th output neurons, respectively. $T$ is the number of neurons. $P(i)$ is the probability of the $i$th output. The SceneNet model is shown in Table 1.

**Table 1.** SceneNet Model.

| Line | Input | Operate | Kernel | Stride |
|------|-------|---------|--------|--------|
| 1 | $128 \times 15 \times 60 \times 1$ | Magconv1 | $1 \times 18 \times 64$ | $1 \times 6$ |
| 2 | $128 \times 15 \times 100 \times 1$ | Gpsconv1 | $1 \times 30 \times 64$ | $1 \times 10$ |
| 3 | $128 \times 15 \times 60 \times 1$ | Lightconv1 | $1 \times 6 \times 64$ | $1 \times 2$ |
| 4 | $128 \times 15 \times 20 \times 1$ | Pressureconv1 | $1 \times 6 \times 64$ | $1 \times 2$ |
| 5 | $128 \times 15 \times 8 \times 64$ | Magconv2 | $1 \times 3 \times 64$ | $1 \times 1$ |
| 6 | $128 \times 15 \times 8 \times 64$ | Gpsconv2 | $1 \times 3 \times 64$ | $1 \times 1$ |
| 7 | $128 \times 15 \times 8 \times 64$ | Lightconv2 | $1 \times 3 \times 64$ | $1 \times 1$ |
| 8 | $128 \times 15 \times 8 \times 64$ | Pressureconv2 | $1 \times 3 \times 64$ | $1 \times 1$ |
| 9 | $128 \times 15 \times 6 \times 64$ | Magconv3 | $1 \times 3 \times 64$ | $1 \times 1$ |
| 10 | $128 \times 15 \times 6 \times 64$ | Gpsconv3 | $1 \times 3 \times 64$ | $1 \times 1$ |
| 11 | $128 \times 15 \times 6 \times 64$ | Lightconv3 | $1 \times 3 \times 64$ | $1 \times 1$ |
| 12 | $128 \times 15 \times 6 \times 64$ | Pressureconv3 | $1 \times 3 \times 64$ | $1 \times 1$ |
| 13 | $128 \times 15 \times 4 \times 4 \times 64$ | Reshape | - | - |
| 14 | $128 \times 15 \times 16 \times 64$ | Mergeconv1 | $1 \times 16 \times 64$ | $1 \times 2$ |
| 15 | $128 \times 15 \times 8 \times 64$ | Mergeconv2 | $1 \times 12 \times 64$ | $1 \times 2$ |
| 16 | $128 \times 15 \times 4 \times 64$ | Mergeconv3 | $1 \times 8 \times 64$ | $1 \times 2$ |
| 17 | $128 \times 15 \times 2 \times 64$ | Flatten | - | - |
| 18 | $128 \times 15 \times 128$ | RNN | - | - |
| 19 | $128 \times 15 \times 200$ | Average | - | - |
| 20 | $128 \times 200$ | Logits | - | - |

Table 1 indicates the process of building SceneNet network, which mainly consists of three convolutional neural networks (CNN), a recurrent neural network (RNN), reshape, flatten and other operations. As shown in Line 1–4, the first part of CNN consists of three convolutional layers with the pre-processed data of magnetometer, GPS, light sensor, and barometer as inputs. In order to extract the intrinsic connections of different data characteristics within a single small window, the second and third part of the CNN are

composed of three convolutional layers based on the first part of the CNN. They accomplish the task of transforming the original signal into abstract features, as shown in Line 5–12. Line 13–20 denote that the abstract features are processed through the reshape layer and CNN for combination first. The previously obtained features are stacked and combined into sequence data as the inputs of RNN. Then, the features for classification are extracted from the sequence data through RNN. After that, a series of classification training processes such as Average and Logits are performed to obtain the final scene classification results.

### 3.4. Triggered Recognition Segment

When the triggered recognition segment is activated, the smartphone camera rotates to collect five images in the forward, left, right, backward, and upward directions. Five images are spliced in the way shown in Figure 8.

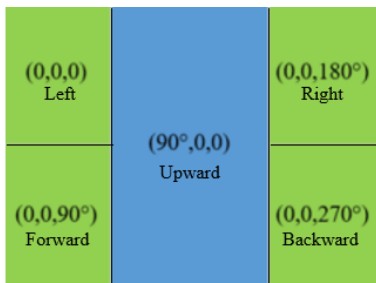

**Figure 8.** The spliced images.

The image is marked with the gimbal angle from which it was taken. The image facing upward is set in the center of the spliced image, which occupies the most significant space. The other images are distributed randomly around the center image. The pixel values of R, G, B channels in the image are adjusted to a distribution with 0 mean and 1 variance, which can improve the generalization performance of the network when training the deep learning network. The spliced images are standardized according to Equation (3),

$$\text{ing\_standardization} = \frac{x_{ij} - \mu}{\max\left(\sigma, \frac{1}{\sqrt{N}}\right)} \quad i \in (1 \dots k) j \in (1..J) \tag{3}$$

where $x_{ij}$ is the $ij$th element of the image matrix, $\mu$ and $\sigma$ are mean and variance of the image, respectively, and $N$ is the number of pixels in the image.

We use histogram information to optimally calculate the mean and variance, where the mean value $\mu$ of each channel of the image is:

$$\mu = \frac{1}{N} \sum_{i=0}^{M-1} iH_i \tag{4}$$

The variance $\sigma$ of each channel of the image is:

$$\sigma^2 = \frac{1}{N} \sum_{i=0}^{M-1} (i - \mu)^2 H_i \tag{5}$$

where $i \in [1 \dots M - 1]$ indicates all possible grayscale values in this image, $M = 256$, $H_i$ is the number of pixels of the current gray value.

Features of the spliced image are extracted and classified by the MobileNet-v2 framework [32] because this framework can be easily integrated into a smartphone and has high accuracy. The three-dimensional output tensor of MobileNet-v2 represents the classification accuracy under the three different scenes.

## 4. Sensor Data Preprocessing and Network Training

We developed an application in the Android system named "Scene Recognition" to assist the environment perception subsystem and decision-making subsystem in the FIAN scheme. In this application, the functions of sensor acquisition frequency setting, data storage, and environmental labeling have been integrated to collect sensor data for network training. The sensor data acquisition process is controlled by the UI interface, as shown in Figure 9, or by the ground station of the MSC flight platform. The IP address communicated with the MAV is preset to 192.168.4.1, and the data acquisition frequency is set to 50 Hz by default.

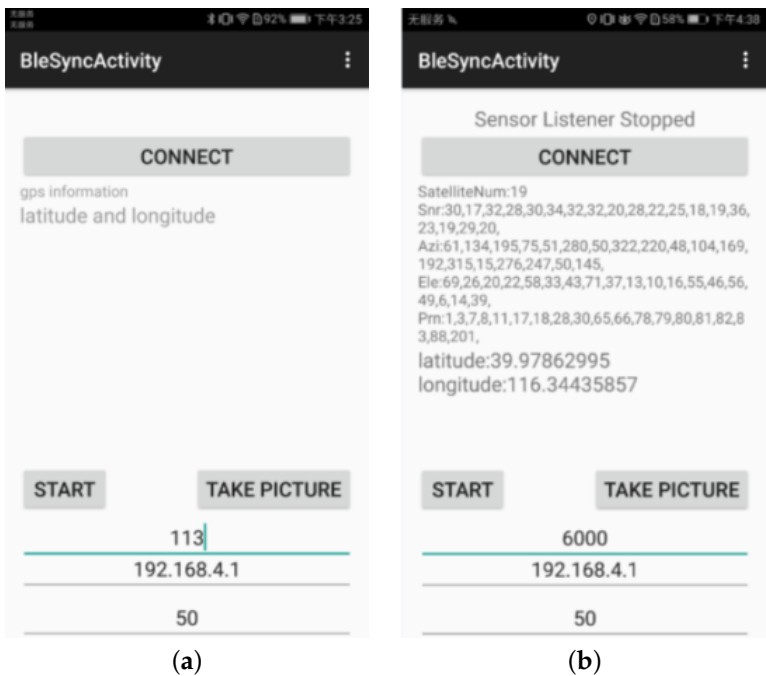

**Figure 9.** The implementation of data collection application. (**a**) The stop interface. (**b**) The collecting interface.

### 4.1. Sensor Data Pre-Processing

The MSC flight platform has carried out more than one thousand and three hundred flight tests under various weather and light conditions in diverse flight scenes. The pre-processing of the raw data from non-vision-based sensors is illustrated in Figure 10.

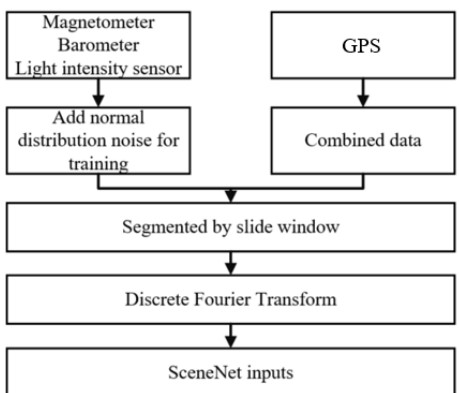

**Figure 10.** The implementation of data collection application.

Firstly, pre-processing of outlier rejection and sliding window-based data cutting are taken for the acquired raw sensor data. In order to improve the tolerance to noise of the

deep learning network, a discrete Fourier transform is performed on the time-series data based on sliding window. Finally, the disposed data are input to the SceneNet for training and making it possible to complete scene classification.

The raw data from the magnetometer, barometer, and light intensity sensor are mixed with unmodeled noise. Normal distribution noises are added to the training process to improve the robustness of the network. The SNR (signal-to-noise ratio) of visible satellites in GPS is combined in below equations.

$$sum = \sum_i SNR_i \tag{6}$$

$$num_{s_1-s_2} = \sum_i F_n(SNR_i) \tag{7}$$

$$F_n(x) = \begin{cases} 1, & s_1 \leq x \leq s_2 \\ 0, & \text{others} \end{cases} \tag{8}$$

$$s_2 = s_1 + 12 \tag{9}$$

$$n = s_1/12 \tag{10}$$

$$s_1 = (0, 12, 24, 36) \tag{11}$$

where $SNR_i$ is the SNR value of the $i$th satellite, $sum$ denotes the sum of SNR, $num_{s_1-s_2}$ indicates the number of satellites with SNR between $s_1$ and $s_2$. GPS data contains five features, including $sum$ and the number of statistical satellites $num_{s_1-s_2}$ in four different SNR intervals. The SNR interval is divided into four parts: 0–12, 12–24, 24–36, 36–48, and the number of satellites in each interval is counted as $num_{0-12}$, $num_{12-24}$, $num_{24-36}$, $num_{36-48}$.

The processed data in each period are separated into 15 intervals by the sliding window ($t = 0.25$ s) and fixed stride ($v = 0.25$ s). In each interval, a Discrete Fourier Transform is applied to convert the processed data into the frequency domain, and $10 \times 2f \times t$ tensors are extracted, where it is the number of magnitude and phase pairs in the frequency domain. In each period, the dimension of the input tensor of SceneNet is $10 \times 2f \times t \times 15$. After all raw data of non-vision sensors from the flight tests are pre-processed, 23,000 sample data with scene labels are collected. These sample data are randomly divided into a training set and a validation set of SceneNet. The ratio of images in the training set and validation set is 4:1.

1300 spliced images with scene labels that serve as the sample data for MobileNet-v2 are randomly divided into the training and test sets. The ratio of numbers of the training and validation set is 4:1. The spliced images with different scene labels are shown in Figure 11.

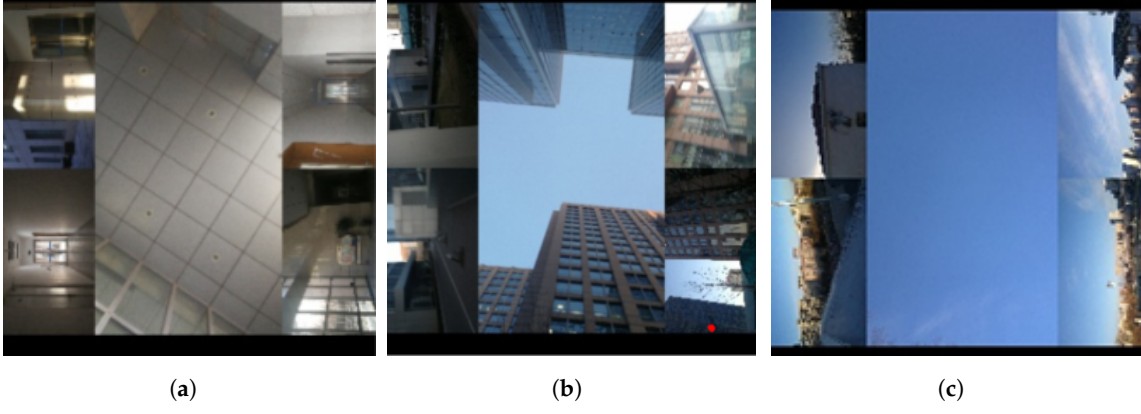

(a)  (b)  (c)

**Figure 11.** The spliced images with scene labels. (**a**) Indoor. (**b**) Intermediate. (**c**) Outdoor.

### 4.2. The Training and Testing Process of SceneNet

An Adam optimizer and cross-entropy loss function are applied in the training process of SceneNet. The value of regularization loss is set to 0.0005, and the value of the initial learning rate is configured to 0.01. The batch size is 128. Tensorflow is used to train the network and record the results of the training process. The classification accuracy and loss value of the training process of SceneNet are shown in Figure 12.

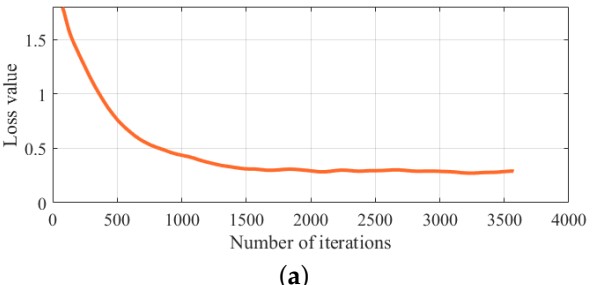

(**a**)

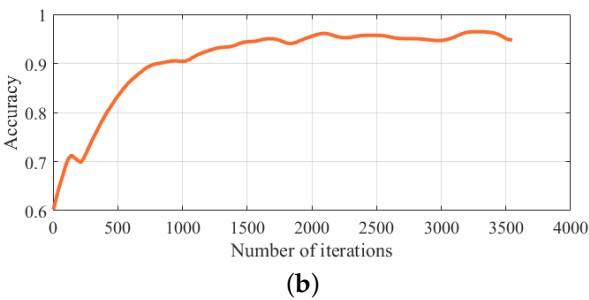

(**b**)

**Figure 12.** The loss function and accuracy in the training process of SceneNet. (**a**) Loss in the training process of SceneNet. (**b**) Accuracy in the training process of SceneNet.

It can be seen that the loss value of the network on the training set gradually decreases, and the classification accuracy gradually increases as the training epochs increase.

To further validate the accuracy of the SceneNet model, it was tested by using a confusion matrix. The confusion matrix of the SceneNet model is illustrated in Table 2. The correct classification rate can reach 99.1% for indoor scenes, 94.9% for intermediate scenes and 98.3% for outdoor scenes. The test results show that the SceneNet model has a high classification accuracy, while the triggered recognition segments can be activated when the classification error is greater than the trigger threshold.

**Table 2.** Confusion matrix.

| Confusion Matrix | | Predicted Label | | |
|---|---|---|---|---|
| | | Indoor | Intermediate | Outdoor |
| True label | indoor | 0.991 | 0.009 | 0 |
| | intermediate | 0.024 | 0.949 | 0.027 |
| | outdoor | 0 | 0.017 | 0.983 |

Furthermore, the performance of SceneNet is compared with a traditional machine learning method PSO-SVM (Particle Swarm Optimization Support Vector Machine) [33]. During the training process of SVM, the RBF kernel function is chosen as the kernel function, and a ten-fold cross-validation method is applied to evaluate the model's performance. PSO with the optimal individual fitness in Figure 13 is used to improve the training speed.

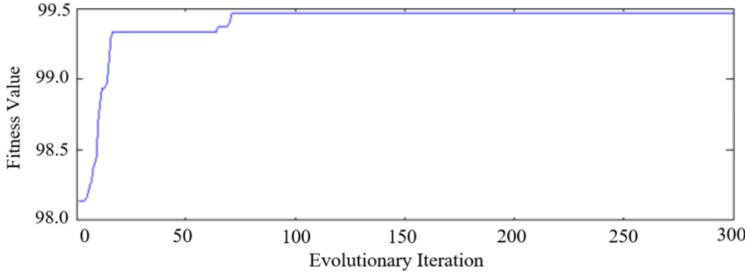

**Figure 13.** The optimal individual fitness value of PSO.

The classification results of PSO-SVM on the validation set are shown in Figure 14a, where the vertical coordinates 0, 1 and 2 represent indoor scenes, intermediate scenes and

outdoor scenes, respectively. The blue line indicates the PSO-SVM predicted category and the red line means the PSO-SVM true category. It can be seen that the main errors occur in intermediate scenes. Figure 14b shows that the classification accuracy based on PSO-SVM scene recognition is 92.3%, and the classification accuracy of SceneNet is 97.4%, which has a 5.1% improvement compared with the PSO-SVM method.

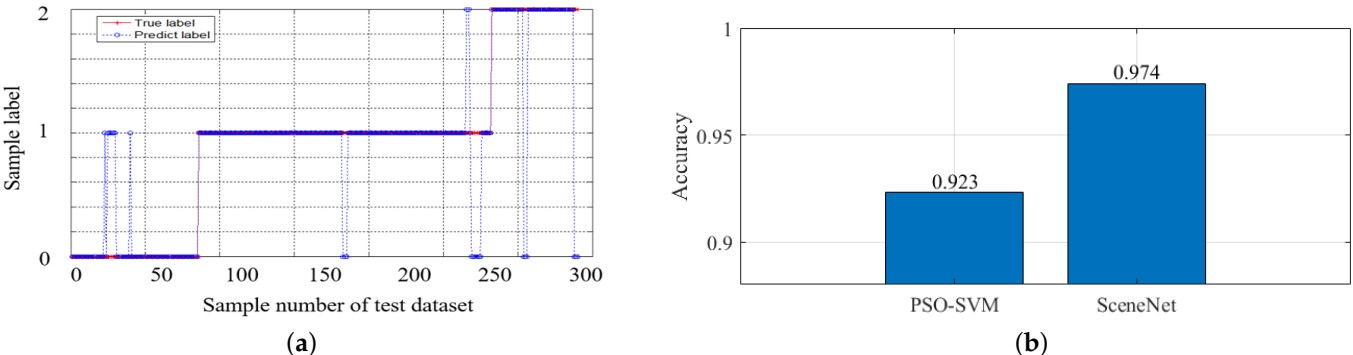

(**a**)                                                                                                   (**b**)

**Figure 14.** Classification result of SceneNet and PSO-SVM. (**a**) The classification result of PSO-SVM. (**b**) The classification accuracy of SceneNet and PSO-SVM.

### 4.3. The Training and Testing Process of MobileNet-v2

For training the MobileNet-v2 based networks, an SGD optimizer and cross-entropy loss function are applied. The model training parameters are as follows: the value of regularization loss is set to 0.00004. The batch size is scheduled to 16, and the moving average factor is placed to 0.9999. Learning rate attenuation is utilized.

Figure 15 denotes that the initial learning rate set and the learning rate changes as the epoch number grows. The loss value of the training process of MobileNet-v2 is shown in Figure 16. It can be seen that the best model of MobileNet-v2 based network achieves 98.2% classification accuracy on the validation set.

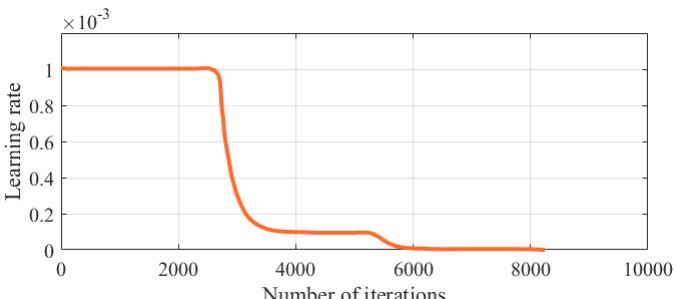

**Figure 15.** Learning rate changes.

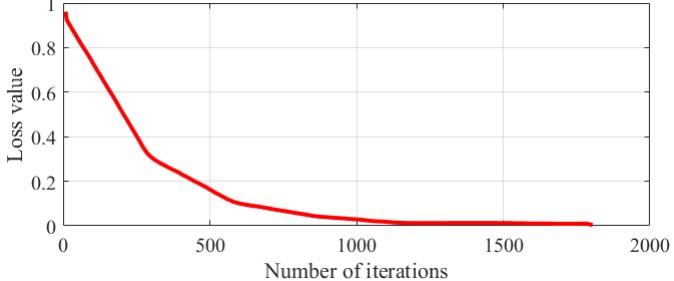

**Figure 16.** Loss value of MobileNet-v2.

Inception-ResNet-v2 [31] is the most effective framework, with 3.7% error on the ILSVRC 2012 validation set. The Inception-ResNet-v2 based network has been trained on

the same training set for comparison. The same training method is adopted for Inception-ResNet-v2 and MobileNet-v2 because they have similar frameworks. The best model of Inception-ResNet-v2 based network achieved 98.5% accuracy on the validation set. However, the MobileNet-v2 based model has lower complexity than that of the Inception-ResNet-v2-based network. Therefore, the Mobilenet-v2 model is more suitable for MAVs.

The performance of scene classification based on the Mobilenet-v2 model is compared with the traditional scene recognition method GISTSVM [34]. For image processing, we first use the GIST feature extraction, and then use the PCA method to reduce the acquired features dimension. After classifying the image scenes, the Mobilenet-v2 uses the PSO method to train the network. The optimal individual fitness values of PSO are shown in Figure 17.

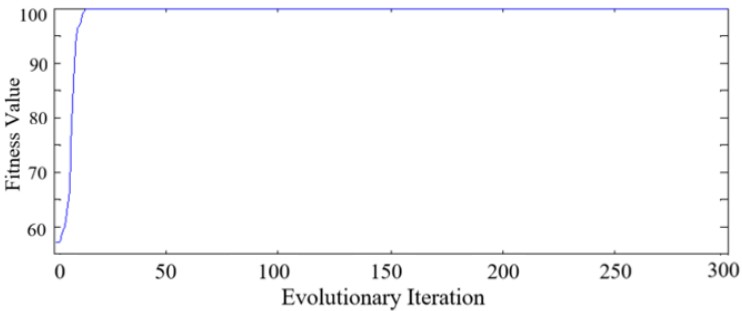

**Figure 17.** The optimal individual fitness value of PSO.

The performance of the classifier on the test set is shown in Figure 18a. GIST-SVM uses the GIST features for classification, the accuracy on the test set is 75.75%. The main errors occur in the intermediate scenes and indoor scenes, which validates that the intermediate scenes are the most inaccurate scenes. Figure 18b shows the scene classification accuracy of the MobileNet-v2 model is improved by 22.5% to 98.2% compared to the GIST-SVM method.

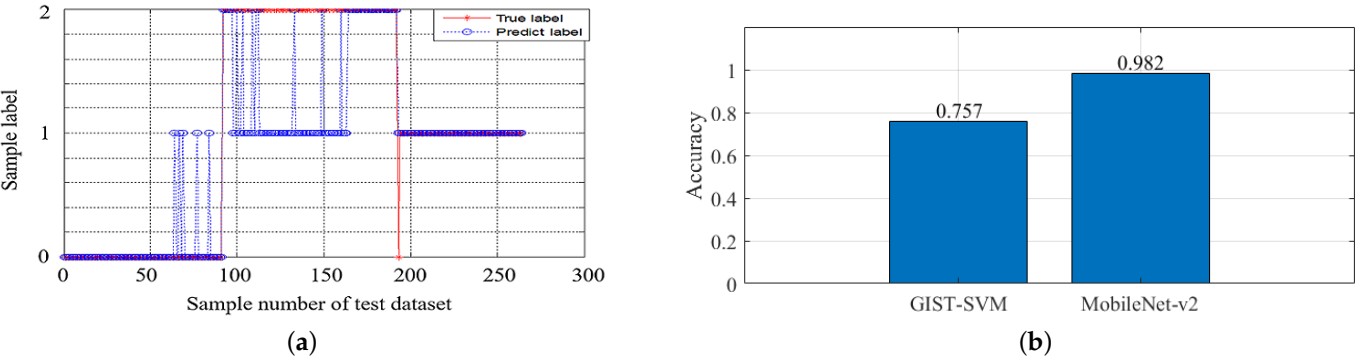

**Figure 18.** Classification result of MobileNet-v2 and GIST-SVM. (**a**) The classification result of GIST-SVM. (**b**) Accuracy of MobileNet-v2 and GIST-SVM.

## 5. Flight Experiment Analysis

The MSC platform flight experiments validate the performance of the TSRM algorithm in the FIAN scheme under diverse environments. The trigger threshold is set to 0.6. The MSC platform carried out two different types of missions during the flight experiments, including single-scene and indoor–intermediate–outdoor flights. The flight scenes of the single-scene flight include a playground, garden, street, building entrance, urban canyon, laboratory, classroom, and corridor at daytime, as well as the street, urban canyon, and corridor at dusk.

Figures 19b, 20b, and 21b demonstrate the scene recognition results of TSRM in day-time garden (outdoor scene), daytime corridor (indoor scene) and dusk building entrance

(intermediate scene), respectively. As can be seen from the Figures 19a and 20a, the confidence probability of the TSRM algorithm for both outdoor and indoor scene recognition during the actual test is above 95%. Figure 21a shows that the minimum scene recognition confidence probability of the TSRM algorithm for intermediate scenes is 58%, which is much higher than the maximum confidence probability of 42% for outdoor scenes. The above analysis shows that TSRM can identify transition scenes quickly and accurately.

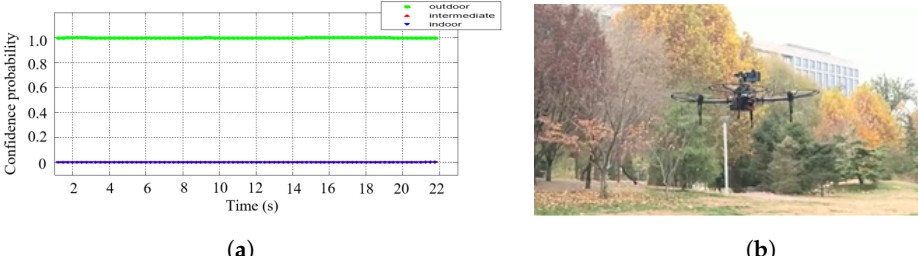

(a)                                          (b)

**Figure 19.** The scene recognition result of flight experiment under garden at daytime.

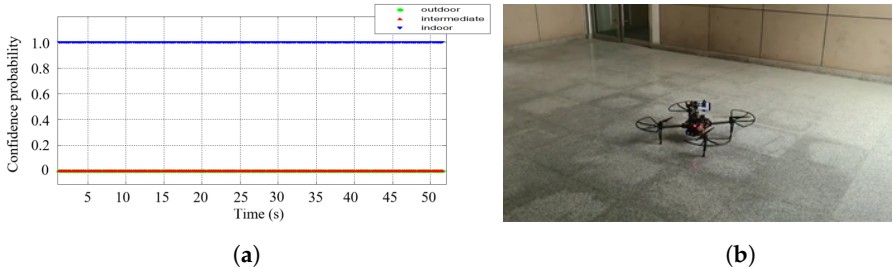

(a)                                          (b)

**Figure 20.** The scene recognition result of flight experiment under corridor at daytime.

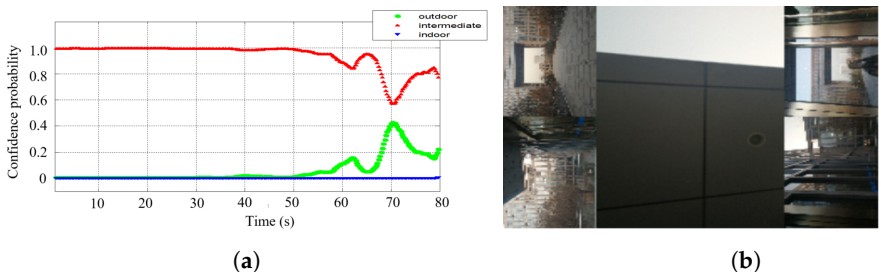

(a)                                          (b)

**Figure 21.** The scene recognition result of flight experiment under building entrance at dusk.

The minimum confidence probability of scene recognition concerning the stitching pictures in other places is shown in Table 3, the probability in red indicates that the triggered recognition segment in the decision-making subsystem is activated.

**Table 3.** Minimum confidence probability.

| Scene | Playground | Street | Urban Canyon | Laboratory | Classroom | Building Entrance |
|---|---|---|---|---|---|---|
| Stitching Pictures | | | | | | |
| Light | Daytime | Daytime | Dusk | Dusk | Daytime | Daytime |
| Minimum confidence probability | 98% | 97.3% | 87.3% | 95.6% | 97.9% | 95.8% |

The MSC platform performed well in all missions and the experimental results are indicated in Table 3, which shows that the confidence probability was above 95.6% in

most scenes. The above data denote that the current scene of the MAV could be accurately distinguished by TSRM. Even under low light conditions such as the dusk in urban canyon, the triggered recognition segment as the second stage of TSRM can be used to achieve confidence probability of 87.3% as shown in red in Table 3.

The execution time and power consumption of the smartphone were estimated when the scene recognition application is activated. As the triggered recognition segment is activated for the TSRM algorithm on Android 7.0, the minimum execution time is 31 ms, and the maximum execution time is 485 ms. The maximum power consumption is about 686 mw. Therefore, the performance of the proposed method meets the requirement of environment adaptive navigation for MAVs.

From the corridor to the garden, the MSC platform flies from the indoor scene, passes the intermediate scene, and finishes at the outdoor scene. As shown in Figure 22, the scene recognition method recognizes the intermediate scene quickly and accurately with a confident probability that increases with time. The triggered recognition segment is activated when the MAV is flying from indoor to intermediate and intermediate to outdoor. The result validates the effectiveness and robustness of the TSRM algorithm.

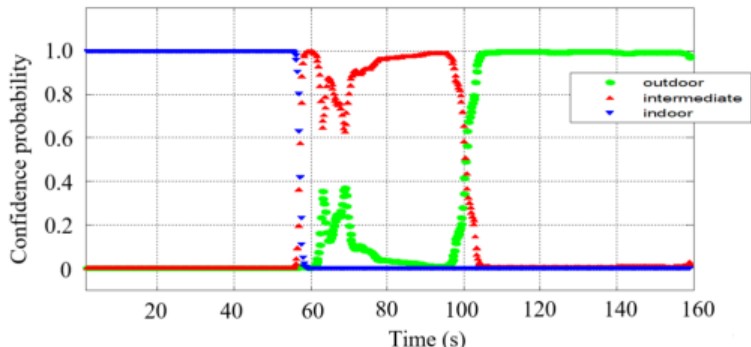

**Figure 22.** The scene recognition result of the actual test in the intermediate scene.

## 6. Conclusions and Future Works

The proposed functional intelligence-adaptive navigation (FIAN) scheme was implemented for MAV scene recognition challenges in diverse environments. In the decision-making subsystem of FIAN, a two-segment scene recognition method consisting of two deep learning frameworks, SceneNet and Mobile Net-V2, can identify the types of scenes with the measurements from environment-sensitive sensors by mimicking the physiological decision-making function of animals. We did some comparative experiments between TSRM and other algorithms. The results show the classification accuracy based on SceneNet has a 5.1% improvement compared with the PSO-SVM method, and the MobileNet-v2 accuracy improves by 22.5% compared to the GIST-SVM method. In addition, the algorithm was loaded into the four-motor MSC platform and tested in a series of indoor–intermediate–outdoor flights. All the experimental results show that the confidence probabilities of scene recognition based on TSRM are above 95.6%. It was verified that the TSRM algorithm has high accuracy and the exploited scheme is very effective in various environment.

In the future, we will attempt to reconfigure various sensors embedded in Pixhawk in complex environments and try to achieve smooth information fusion. As an interesting and important research topic, we will also consider extending the environmental adaptive scene recognition experiments proposed in this paper to other complex environments, such as forests and caves, based on the similarity of scene features.

**Author Contributions:** L.W. and L.F. made major contributions to this paper together. They exploited the FIAN scheme and MSC platform, applied it to the real drone and wrote the paper together. Y.L. provided important technical advice on the formula derivation and the paper revision. Y.W. was responsible for the derivation of the formula and simulation process. N.T. took responsibility for the theoretical research and analysis of the FIAN scheme. All authors have read and agreed to the published version of the manuscript.

**Funding:** This work was supported in part by grant from the National Natural Science Foundation of China (No. 61773037) and Beihang Virtual Simulation First-class Course Project (No. 42020200).

**Institutional Review Board Statement:** Not applicable.

**Informed Consent Statement:** Not applicable.

**Data Availability Statement:** The data that support the findings of this study are available from the corresponding author upon reasonable request.

**Conflicts of Interest:** The authors declare no conflict of interest.

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
