# Peer review of "Functional Intelligence-Based Scene Recognition Scheme for MAV Environment-Adaptive Navigation"

_drones, doi:10.3390/drones6050120_

Round 1

Reviewer 1 Report

In my opinion, the main contributions of paper include: (1) A functional intelligence-adaptive navigation scheme was proposed to improve the environment-adaptive capacity of MAVs navigation. (2) Inspired by the hunting behavior of owls, two deep learning frameworks of SceneNet and Mobile Net-V2 are used to construct a two-level scene recognition method  to accurately identify the current flight scene. All the experiments were evaluated on the four-rotor MAV-Smartphone combined (MSC) platform.

(1).The references are sufficient. However, some related paper from Drones should be referenced.

(2). The authors should provide a brief explanation of Table 1 and Figure 10.

(3). The Figure 11 is too large and incongruous.

(4).The manuscript contains some mistakes in grammar and spelling. Please check it carefully.

Reviewer 2 Report

The current manuscript is devoted to investigate a functinal inteligence-based scence for the MAV environment-adaptive navigation. 

It can be accepted after some revisions. 

1) What is 'Bioinspired decision-making system' in the keywords? The author did not mention it in the abstract? Why and in the manuscript, the author also do not mention it. So this keyword is not correct. 

2) Need to improve the figure quality to show the information and text in a clear way. 

3) Except for the urban building area, How this system use in the forest area? The author can add some more results for it or discussions. 

4) Abstract and conclusion need to be re-written to including more key results in this manuscript. 

Reviewer 3 Report

The paper “Functional Intelligence-based Scene Recognition Scheme for MAV Environment-adaptive Navigation” proposes a scheme for environment classification problem using two deep learning framework. The topic of the work will be interesting for the specialist in MAV navigation. The paper is of high-quality, it is well written and clear. It can be accepted for the publication after a few minor corrections.

In Fig. 3 the measurements of magnetometer cannot be in nT, it is too low value for Earth environment. It seems that it should be in µT. Also, the axis directions should be defined in MAV-fixed reference frame.

It is not quite clear explained what is the trigger function of ‘Belief’ in Fig.6 for decision making process. More comments are required.

A short explanation of meaning of the content of Table 1 with SceneNet Model will improve the text.

What is the purpose of the spliced image standardization according to the eq. (3)? It seems that the images are color. Then how the mean and variance are calculated?

Axes labels for Fig. 12, 14b, 15, 16, 18b should be provided. Also the legend for Fig.16 with two curves is required.

Round 2

Reviewer 2 Report

Accepted as it is